# Generation of Hydrogen Peroxide and Downstream Protein Kinase D1 Signaling Is a Common Feature of Inducers of Pancreatic Acinar-to-Ductal Metaplasia

**DOI:** 10.3390/antiox11010137

**Published:** 2022-01-08

**Authors:** Heike R. Döppler, Geou-Yarh Liou, Peter Storz

**Affiliations:** 1Department of Cancer Biology, Mayo Clinic, Jacksonville, FL 32224, USA; doeppler.heike@mayo.edu (H.R.D.); gliou@cau.edu (G.-Y.L.); 2Department of Biological Sciences, Center for Cancer Research & Therapeutic Development, Clark Atlanta University, Atlanta, GA 30314, USA

**Keywords:** hydrogen peroxide, oxidative stress, ADM, acinar-to-ductal metaplasia, pancreas, PKD, Protein Kinase D

## Abstract

Pancreatic acinar-to-ductal metaplasia (ADM) is a reversible process that occurs after pancreatic injury, but becomes permanent and leads to pancreatic lesions in the presence of an oncogenic mutation in KRAS,. While inflammatory macrophage-secreted chemokines, growth factors that activate epidermal growth factor receptor (EGFR) and oncogenic KRAS have been implicated in the induction of ADM, it is currently unclear whether a common underlying signaling mechanism exists that drives this process. In this study, we show that different inducers of ADM increase levels of hydrogen peroxide, most likely generated at the mitochondria, and upregulate the expression of Protein Kinase D1 (PKD1), a kinase that can be activated by hydrogen peroxide. PKD1 expression in acinar cells affects their survival and mediates ADM, which is in part due to the PKD1 target NF-κB. Overall, our data implicate ROS-PKD1 signaling as a common feature of different inducers of pancreatic ADM.

## 1. Introduction

In response to inflammation, growth factor signaling or oncogenic signaling, pancreatic acinar cells can undergo a dedifferentiation process named acinar-to-ductal metaplasia (ADM) [1]. ADM is characterized by the transcriptional downregulation of acinar cell markers such as amylase and carboxypeptidase A [2], and increased expression of ductal markers, such as cytokeratin 19 (CK19) and mucin-1 [3]. Cells that undergo ADM are in a less-differentiated, duct-like state [1,4,5], and feature the capacity for pancreatic regeneration. However, ADM cells can be locked into their dedifferentiated state when expressing mutant (active) KRAS. This is an irreversible process, leading to development of low-grade pancreatic lesions, as long as KRAS is active [1,6]. Eventually, low-grade lesions can acquire additional oncogenic mutations and progress to carcinoma in situ and pancreatic cancer [1,5]. Thus, understanding the signaling processes that drive ADM could provide insight into how the pancreas regenerates after a damaging insult, as well as an understanding of the earliest events driving tumor formation.

The signaling mechanisms that regulate ADM can be studied using a three-dimensional (3D) ex vivo organoid cell culture model. In this model, mouse primary acinar cell clusters are isolated from the pancreas, embedded in collagen I matrix and transdifferentiated in the presence of growth factors that activate EGFR (i.e., TGFα and EGF), inflammatory macrophage (IM)-secreted molecules (i.e., TNF, RANTES/CCL5), oncogenic KRAS, or other stimuli [7,8,9,10,11]. These ex vivo studies reliably correlate with animal studies. For example, transgenic animal models in which either mutant KRAS [12] or TGFα [13] is expressed in the acinar cells of the pancreas induce acinar cell metaplasia and further progression. Moreover, the induction of pancreatic inflammation with caerulein, a cholecystokinin analog, showed the importance of inflammatory macrophage-secreted factors as drivers of ADM [10]. In this study, we employed this ex vivo 3D cell culture system to investigate the open question of whether different activators of ADM can act though a common signaling pathway.

Using conditional knockout mice, we previously demonstrated that the serine/threonine kinase PKD1 (but not the closely related kinases PKD2 and PKD3) is a critical component of TGFα- and KRAS-driven ADM [14]. Both inducers increase PKD1 expression in vivo and ex vivo in 3D cell culture [14]. Moreover, oncogenic KRAS was shown to induce mitochondrial oxidative stress [15], which is a potent activator of PKD1 [16,17,18]. Downstream of KRAS, PKD1 has been shown to activate transcriptional regulation via activation of nuclear factor-κB (NF-κB) and Notch [14,15], both key factors driving the ADM process [14,19]. The aim of the current study was to determine whether the generation of hydrogen peroxide is a critical factor that is common to different activators of ADM. Another aim was to determine whether H_2_O_2_ drives the dedifferentiation process to duct-like cells through the kinase PKD1.

## 2. Materials and Methods

### 2.1. Cells, Antibodies and Reagents

The raw 264.7 macrophage cells (ATCC, Manassas, VA, USA) were maintained in DMEM (high glucose) media containing 10% FBS and 100 U/mL penicillin/streptomycin in a 37 °C incubator supplemented with 5% CO_2_. To obtain the conditioned media, 3 × 10^6^ cells per 10 cm dish were grown in media used for acinar cell 3D collagen explant culture (Waymouth’s media plus supplements, see below) and supernatants were collected. The conditioned media was freshly prepared for each experiment. The PKD1 antibody (AP06569PU-N) was from Acris (San Diego, CA, USA), the β-actin (A5441) antibody was from Sigma-Aldrich (St. Louis, MO, USA). The secondary HRP-linked anti-mouse or anti-rabbit antibodies were from Jackson ImmunoResearch (West Grove, PA, USA). The recombinant murine CCL5/RANTES and TNFα were purchased from PeproTech (Rocky Hill, NJ, USA). The recombinant TGFα was from R&D Systems (Minneapolis, MN, USA). The hydrogen peroxide was from Thermo Fisher (Rochester, MN, USA) and the EUK134 was from Cayman Chemical (Ann Arbor, MI, USA). The PKD-specific inhibitor kb-NB-142-70 was described previously [14,20]. The collagenase I was from Millipore/Sigma (St. Louis, MO, USA). The rat tail collagen I was from BD Biosciences (San Diego, CA, USA).

### 2.2. Viral Vectors and Viral Transduction

The adenovirus to express catalase (rAD-CVM-Cat), the mitochondria-targeted catalase (rAD-CVM-mCat) and the control, empty adenovirus (rAD-Null), were obtained through the University of Iowa Gene Transfer Vector Core. The adenovirus to express super-dominant IκBα (IκBα.SD, IκBα.S32A.S36A) was purchased from Vector Biolabs (Philadelphia, PA, USA). The vectors for the lentiviral expression of PKD1 alleles pLenti6.3/V5-GFP-PKD1.CA and pLenti6.3/V5-GFP-PKD1.KD were described previously [14]. The lentiviral plasmids to knock down murine PKD1 were purchased from Sigma-Aldrich. As described in our previous work [14], two hairpin sequences (mPKD1-shRNA#1 and mPKD1-shRNA#2) showed almost identical efficiency and a specific knockdown of mouse PKD1. Here, we used mPKD1-shRNA#1 with the following sequence: 5′-CCGGGAGTGTTTGTTGTTATGGAAACTCGAGTTTCCATAACAACAAACACTCTTTTT-3′. The lentiviral plasmids to knock down murine p22^phox^ were purchased from Sigma-Aldrich and are described in [15]. The adenoviral and lentiviral transductions were performed as described in previous research [14,21]. In all the experiments, the virus was used at 10^7^ ifu/mL.

### 2.3. Proliferation Assays

The cells were isolated and infected as indicated. The cells were plated and their DNA content was measured using the CyQUANT Cell Proliferation Assay kit (Invitrogen, Waltham, MA, USA) on a Synergy HT plate reader (BioTek, Winooski, VT, USA).

### 2.4. Isolation of Primary Peritoneal Macrophages

The mice were injected with 2 mL of 5% aged thioglycollate. Five days later, the mice were euthanized and the peritoneal macrophages were collected via a single injection of 10 mL RPMI-1640 plus 10% FBS into the peritoneal cavity and subsequent withdrawal (described in detail in [10,22]). The peritoneal exudate was centrifuged (233× *g*) and washed with RPMI-1640 media plus 10% FBS before plating onto 10 cm tissue culture dishes. After one hour at 37 °C, the plates were washed 2–3 times to remove non-adherent cells, and fresh media was added. The conditioned media was prepared as described above for the Raw 264.7 cells. These experiments were conducted under a Mayo Clinic IACUC-approved protocol (A00004882) and were in accordance with institutional guidelines and regulations.

### 2.5. Isolation of Primary Pancreatic Acinar Cells

The primary pancreatic acinar cells were isolated from the C57BL/6J mice using a protocol that was described in detail previously [11,14,21]. In brief, the pancreas was washed twice with ice-cold HBSS media, minced into 1 to 5 mm pieces and digested with 5 mL of 2 mg/mL collagenase I in HBSS (37 °C, shaker). The digestion was terminated by the addition of an equal volume of ice-cold HBSS media with 5% FBS. The digested pieces were washed twice (HBSS plus 5% FBS) and pipetted through 500 µm and 105 µm meshes. The acinar cell suspension was added dropwise to 20 mL The HBSS media was supplemented with 30% FBS. The acinar cells were pelleted (233× *g*, 2 min, 4 °C) and re-suspended in 10 mL Waymouth’s complete media (1% FBS, 0.1 mg/mL trypsin inhibitor, 1 µg/mL dexamethasone). The isolation of the acinar cells was conducted using excess mice and the procedures were performed after the animals were euthanized. All the experiments were in accordance with IACUC regulations.

### 2.6. Acinar-to-Ductal Metaplasia Assay

This method is demonstrated in detail in [21]. In short, freshly isolated primary pancreatic acinar cells were embedded in collagen I/Waymouth’s media (w/o supplements). Waymouth’s complete media was added on top of the cell/gel mixture and replaced every other day. Inhibitors or compounds were added at indicated concentrations to both the cell/gel mixture and the media on top. For the viral transductions, the acinar cells were infected for 3–5 h before embedding in the collagen I/Waymouth’s media mixture. If not indicated otherwise, the number of ducts per well was determined at day 5, and photographs were taken to document the cellular structures.

### 2.7. Measurement of ROS Generation

The primary pancreatic acinar cells were labeled with H_2_DFFDA (20 µM) in phenol red-free Waymouth’s complete media at 37 °C for 20 min. The cells then were washed with HBSS, transferred to phenol red-free Waymouth’s media and stimulated as indicated. ROS generation (formation of fluorescent DCF) over indicated time was determined at 495/529 nm (excitation/emission) using a SpectraMax M5 fluorescent microplate reader (Molecular Devices, Sunnyvale, CA, USA).

### 2.8. Quantitative PCR

The RNA was extracted using an RNeasy Plus Kit (Qiagen, Germantown, MD, USA), and converted to cDNA using a High Capacity cDNA RT Kit (Applied Biosystems, Foster City, CA, USA). Each qPCR reaction used Taqman Fast 2x PCR Mix (Applied Biosystems) along with mGapdh (Mm99999915_g1), mPrkd1 (Mm00435790_m1), mCyba (p22^phox^; Mm00514478_m1), mPdx1 (Mm00435565_m1), mKrt19 (Mm00492980_m1), mHes1, (Mm01342805_m1) or mMuc1 (Mm00449604_m1) primer/probe sets (Applied Biosystems). The reactions were run on a QuantStudio 7 Flex Real-Time PCR System (Applied Biosystems). All the C_T_ values were normalized to Gapdh and the ΔΔC_T_ method was used to calculate the fold changes.

### 2.9. Cell Lysis and Western Blot

The cells were washed twice using ice-cold PBS (140 mM NaCl, 2.7 mM KCl, 8 mM Na2HPO4, and 1.5 mM KH2PO4 [pH 7.2]) and lysed using buffer A (50 mM Tris-HCl, pH 7.4, 1% Triton X-100, 150 mM NaCl, 5 mM EDTA, pH 7.4) plus a protease inhibitor cocktail (Sigma-Aldrich, St. Louis, MO, USA). After a 30 min incubation on ice, the lysates were centrifuged (16,200× *g*, 4 °C, 15 min), the supernatants were collected and the protein concentration was measured using a BioRad Protein Assay (BioRad, Hercules, CA, USA). Following SDS-PAGE (10% gel; 30 mA) and transfer to nitrocellulose membrane, proteins of interest were detected via Western blot using the indicated primary antibodies at 1:2000 and horseradish peroxidase (HRP)-conjugated secondary antibodies at 1:5000.

### 2.10. Quantification and Statistical Analysis

All the cell biological and biochemical experiments were performed independently of each other at least three times. For the ADM assays, three biological replicates were performed using pancreata from three different mice. The quantification of the ductal area was performed using ImageJ. The data, when presented as bar graphs, show individual values (dots), mean and ± standard deviation (SD). If not stated otherwise in the figure legends, the *p* values were acquired with the unpaired Student’s *t*-test with Welch’s correction using Graph Pad software (GraphPad Inc., La Jolla, CA, USA). The *p* values are included in the graphs and *p* < 0.05 was considered statistically significant.

## 3. Results

### 3.1. TGFα Induces Pancreatic Acinar-to-Ductal Metaplasia through Hydrogen Peroxide

The activation of EGFR signaling through its ligands, EGF and TGFα, in the acinar cells induces their metaplasia to a duct-like phenotype [23], but the underlying cellular signaling is not well understood. The signaling mechanisms driving ADM can be investigated in acinar organoid culture [24]. When seeded in three-dimensional (3D) collagen culture, isolated murine primary acinar cell clusters undergo ADM after treatment with TGFα (Figure 1A). This is accompanied by the upregulation of the mRNA expression of cytokeratin 19 (*Krt19*), pancreatic and duodenal homeobox 1 (*Pdx1*), glycoprotein mucin 1 (*Muc1*), and the Notch target gene Hes1 (*Hes1*), all typical markers that indicate the transdifferentiation process (Appendix A). We noticed that the treatment of isolated acinar cells with TGFα leads to a slight but steady and significant increase in reactive oxygen species over time (Figure 1B). Such TGFα-generated oxidative stress is important for driving the ADM process, since acinar cell organoids that are treated with TGFα in the presence of the antioxidant N-Acetyl-L-Cysteine (NAC), or with EUK134, a synthetic salen-manganese complex with SOD and catalase activity [25], completely fail to undergo transdifferentiation (Appendix A and Figure 1C). Similar inhibitory effects on TGFα-induced ADM are observed when acinar cells are transduced with catalase, suggesting that hydrogen peroxide is the oxidative stress component that drives the transdifferentiation process (Figure 1D).

### 3.2. Macrophage-Secreted Factors Also Induce Pancreatic ADM through Hydrogen Peroxide

We next tested whether oxidative stress is a common feature of how different factors drive ADM. We previously showed that macrophage-conditioned media can initiate the ADM process ([10] and Appendix A). Similar to what was observed for TGFα (Figure 1B), macrophage-conditioned media also increased cellular oxidative stress (Figure 2A). Moreover, either treatment with EUK134 (Figure 2B) or the transduction of catalase significantly decreased macrophage-conditioned media-induced ADM events (Figure 2C), and ductal area (Figure 2D).

We previously identified the cytokines/chemokines CCL5/RANTES and TNFα as the major factors that drive ADM in macrophage-conditioned media [10], with CCL5 being the more potent inducer (Figure 2D). Both factors increased cellular oxidative stress (Figure 2F). Moreover, as seen for the macrophage-conditioned media, the induction of ADM by both of these factors was efficiently blocked by the transduction of catalase (Figure 2G,H). Overall, these data suggest that the induction of oxidative stress is a common feature downstream of the different inducers of ADM, and the generation of hydrogen peroxide may be a driver of the process.

### 3.3. Mitochondria-Generated Hydrogen Peroxide Can Drive Acinar-to-Ductal Metaplasia

We next tested whether hydrogen peroxide can drive ADM and whether it stems from the NADH/NADPH oxidase complex or from mitochondria. The treatment of acinar organoids with hydrogen peroxide induced ADM approximately twofold (Figure 3A), although the ducts formed were smaller in size when compared to TGFα (Figure 3B). We then tested whether TGFα induces ADM through the hydrogen peroxide that is produced by the NADH/NADPH oxidase complex. A knockdown of p22^phox^ (Appendix A), which is a critical component of each of the different NADH/NADPH oxidase complexes, did not affect TGFα-mediated ADM (Appendix A). By contrast, the expression of mitochondria-targeted catalase (mitoCat) via adenoviral transduction completely blocked TGFα-mediated ADM, suggesting the mitochondrial electron transport chain as a source for hydrogen peroxide driving the transdifferentiation process (Figure 3C). Similarly, ADM induced by M-CM was blocked when a mitochondria-targeted catalase was expressed (Figure 3D), but not when p22^phox^ was knocked down (Appendix A).

### 3.4. Hydrogen Peroxide Drives ADM via Protein Kinase D1

PKD1 has been shown to be activated by hydrogen peroxide [17,18]. Moreover, we previously showed that PKD1 is a key driver of ADM downstream of EGFR and Kras^G12D^ signaling in vitro and in vivo [14,15]. Therefore, we next tested whether PKD1 also acts downstream of M-CM and CCL5, making it a common feature needed for the ADM process. In acinar cells, M-CM increased PKD1 levels (Figure 4A) and a knockdown of PKD1, using a previously published specific shRNA [14], decreased the number of ADM events (Figure 4B, Appendix A). Moreover, CCL5, the major factor in M-CM that drives ADM, also induced PKD1 expression at mRNA and protein levels (Appendix A, Figure 4C). As observed for M-CM, a knockdown of PKD1 not only decreased the number of CCL5 induced ADM events (Figure 4D), but also the area of ductal structures (Appendix A). Similar effects of PKD1 knockdown have been demonstrated for TGFα and mutant KRAS [14].

Since hydrogen peroxide is downstream of the inducers of ADM, we then tested whether the induction of ADM via hydrogen peroxide is inhibited when PKD1 activation is blocked. The treatment of acinar cell organoids with the PKD-specific inhibitor kb-NB-142-70 showed that active PKD drives ADM downstream of hydrogen peroxide (Figure 4E). Similarly, the lentiviral transduction of a constitutively-active (PKD1.S738E.S742E; PKD1.CA), but not a kinase-dead (PKD1.K612W; PKD1.KD) version of PKD1 into acinar cells led to an increase in ADM events (Figure 4F). Overall, these data suggest the upregulation and ROS-mediated activation of PKD1 as common features essential for the ADM process.

### 3.5. PKD1 Drives ADM through Canonical NF-κB Signaling

We next investigated whether the expression of PKD1 contributes to acinar cell survival. Isolated primary acinar cells, when seeded without surrounding extracellular matrix, undergo apoptosis within a few days [26]. We found that the lentiviral expression of PKD1 in isolated acinar cells can increase acinar cell survival (Figure 5A). Survival downstream of PKD1 is regulated via signaling to one of its main target pathways, canonical NF-κB [18]. NF-kB is also one of the transcription factors that drives ADM [4,15]. Therefore, we tested whether the activation of NF-κB through PKD1 drives ADM in pancreatic acinar cells in 3D explant organoid culture. We found that PKD1-induced ADM can be partially blocked after the adenoviral transduction of a superdominant IκBα (IκBα.SD), which is mutated to block canonical NF-κB activation (Figure 5B). Similarly, the expression of IκBα.SD blocked ADM induced by the PKD1 upstream activators TGFα, CCL5 (both Appendix A), TNF, MCM (both [10]) and mutant KRAS [15]. This suggests that PKD1 drives ADM not only via the previously reported effects on Notch signaling [14], but also though NF-κB. In summary, we show here that ROS-PKD1-NF-κB signaling is downstream of common inducers of acinar cell metaplasia and is a critical driver of the transdifferentiation process.

## 4. Discussion

Pancreatic acinar-to-ductal metaplasia occurs in response to inflammation, as well as growth factor and KRAS signaling. ADM is a reversible process and is involved in pancreatic regeneration when pancreatitis resolves [1]. It was shown in different mouse models that in the presence of an oncogenic KRAS mutation, ADM is irreversible and leads to the development of pancreatic intraepithelial neoplasia (PanIN), which can further develop into pancreatic ductal adenocarcinoma (PDAC) [4,5,12,27]. While the factors that induce ADM are increasingly identified, it remains unclear whether they utilize a common signaling pathway to promote their effects on acinar cell plasticity.

In this study, we show that the generation of hydrogen peroxide is an event that occurs downstream of all the tested inducers of ADM (Figure 1 and Figure 2). Moreover, ADM can be induced via extrinsic hydrogen peroxide (Figure 3A,B). We also show that mitochondrially-targeted catalase and not inhibition of NADPH oxidase can block ADM, suggesting that mitochondria are the origin of ROS signaling (Figure 3C,D; Appendix A and [15]). This seems to be different to fully developed pancreatic cancer, where KRAS-induced NADPH oxidase signaling increases cellular superoxide and hydrogen peroxide levels [28]. In our previous work, we showed that during the ADM process, mutant KRAS induces mitochondrial dysfunction, which results in increased mitochondrial oxidative stress [15,29]. Our data presented here expand this finding to other inducers of ADM, such as TGFα and M-CM. Protein Kinase D1 (PKD1) can be activated by mitochondrial oxidative stress [16,17,18]. During the ADM process, PKD1 is activated downstream of EGFR and mutant KRAS [14], and here we show that it is also downstream of M-CM and CCL5 (Figure 4). Moreover, extrinsic hydrogen peroxide-induced ADM was efficiently blocked when PKD1 was inhibited (Figure 4E).

Pancreatic acinar-to-ductal metaplasia was previously linked to the activation of the transcription factors Notch [30,31], NF-κB [10], STAT3 [32] and NFAT1/4 [33,34], all of which are directly or indirectly activated by PKD1 in acinar cells. For example, it was shown that downstream of KRAS signaling (mutant KRAS and EGFR-KRAS), PKD1 regulates ADM via the activation of Notch [14]. This is most likely achieved through the activation of MMP7, which can cleave Notch to generate the activated version NICD (Notch intracellular domain) [31]. Consequently, Notch target genes such as *Hey1*, *Hes1*, *Sox9* and pancreatic and duodenal homeobox-1 (*Pdx1*), which are increasingly expressed in cells that undergo ADM [31,35,36,37], are all induced by active PKD1 [14]. Another target transcription factor for PKD1 is NF-κB, which in acinar cells upregulates the expression of EGFR and its ligands [15] and possibly contributes to signal amplification via EGFR signaling to induce STAT3 and NFAT1/4 [33,34]. STAT3 has been shown to regulate ADM downstream of YAP1 [1,32], which is also regulated via PKD1 [38]. However, it should be noted that other factors, such as Myc and KLF4, have been shown to be required for the ADM process in mice [39,40] and that the relation and role of PKD1 in their activation is not yet established.

A pathological progression model for PDAC suggests that it originates in ADM progenitor cells that give rise to PanIN [1,5,41]. Since our data show that ROS-PKD1 signaling is a central driver of the ADM process, the pharmacologic inhibition of this signaling pathway may be an effective strategy to prevent early oncogenic changes. Options to reduce ADM events in vivo could include the use of antioxidants or PKD1 inhibitors. While antioxidant strategies to target ROS homeostasis have been tested for PDAC with mixed results (discussed in [42]), the inhibition of PKD1 as a downstream target for ROS may be a more-specific option. However, at this point, no PKD inhibitors have been developed that can be clinically used. Further, inhibitor specificity for the PKD1 isoform may be important, since other PKD isoforms, such as PKD3, can regulate critical acinar cell functions, including amylase secretion [43].

## 5. Conclusions

Pancreatic acinar-to-ductal metaplasia is a de-differentiation process and cells that undergo ADM can initiate the formation of pancreatic precancerous lesions when an oncogenic mutation of KRAS is present (reviewed in [1,5]). Multiple inducers of ADM have been identified [7,8,9,10,11], but so far, no common underlying signaling mechanism has been described. In this study, we identify the generation of mitochondrial ROS and ROS-mediated activation of PKD1 as a common signaling pathway for the inducers of ADM, including CCL5, TNF, EGFR and KRAS. Since PKD1 links to most transcription factors crucial for the acinar cell de-differentiation process, this kinase may be a critical and targetable component to develop strategies to prevent the development of pancreatic cancer.

## Figures and Tables

**Figure 1 antioxidants-11-00137-f001:**
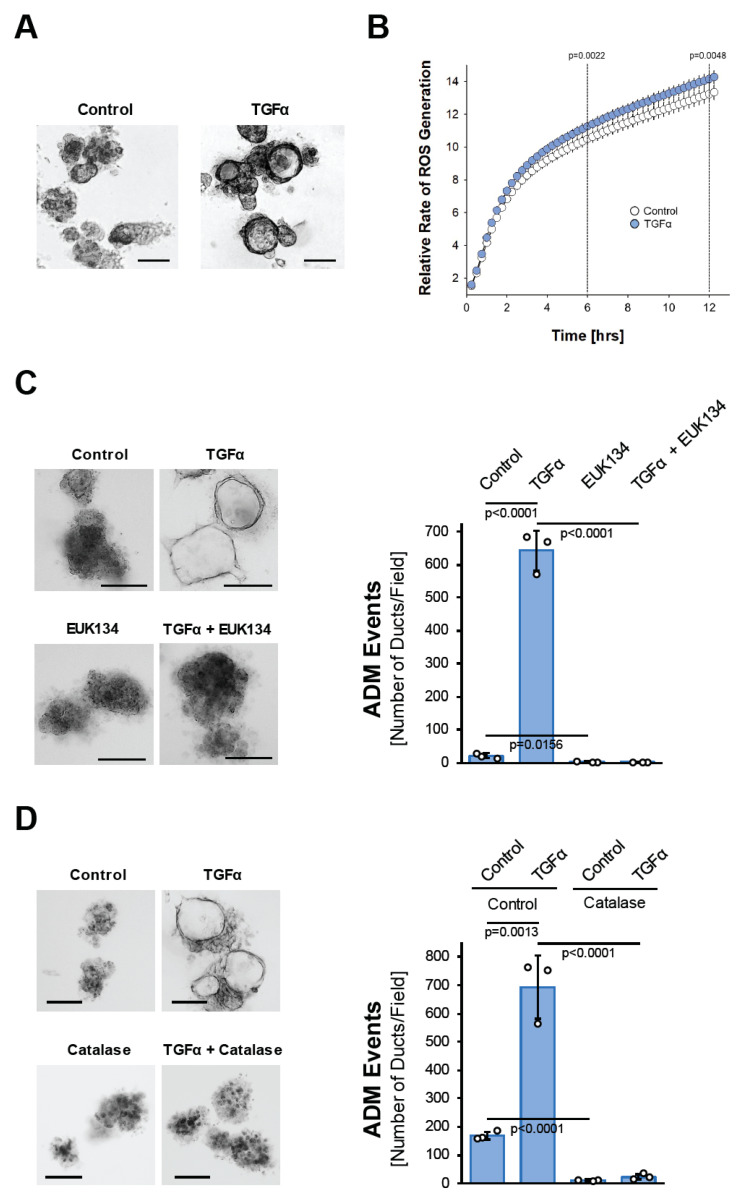
TGFα induces pancreatic acinar-to-ductal metaplasia through hydrogen peroxide. (**A**): Mouse pancreatic acinar cells were isolated and seeded in 3D collagen culture. ADM was induced with TGFα (50 ng/mL) and ducts formed were photographed. The bar indicates 100 µm. (**B**): Primary mouse acinar cells were stimulated as indicated and labeled with H2DFFDA. Generation of intracellular ROS (fluorescent DCF) was measured over the indicated time period. (**C**): Mouse pancreatic acinar cells were isolated and seeded in 3D collagen culture. As indicated, cells were treated with EUK134 (25 µM) and ADM was induced with TGFα (50 ng/mL). Ducts formed were photographed (left side; bar = 100 µm) and quantified (right side) as indicated in the Methods section. (**D**): Mouse pancreatic acinar cells were isolated, infected with empty adenovirus (rAD-Null) or adenovirus to express catalase (rAD-CVM-Cat) and seeded in 3D collagen culture. As indicated, ADM was induced with TGFα (50 ng/mL). Ducts formed were photographed (left side; bar = 100 µm) and quantified (right side) as indicated in the Methods section. (**A**–**D**): All experiments shown were performed in triplicates (except (**B**); n = 6) for at least three times and obtained similar results in each repeat. Statistical analysis between two groups was performed using the Student’s *t*-test. A *p* value of 0.05 was considered statistically significant and values are included in the graphs.

**Figure 2 antioxidants-11-00137-f002:**
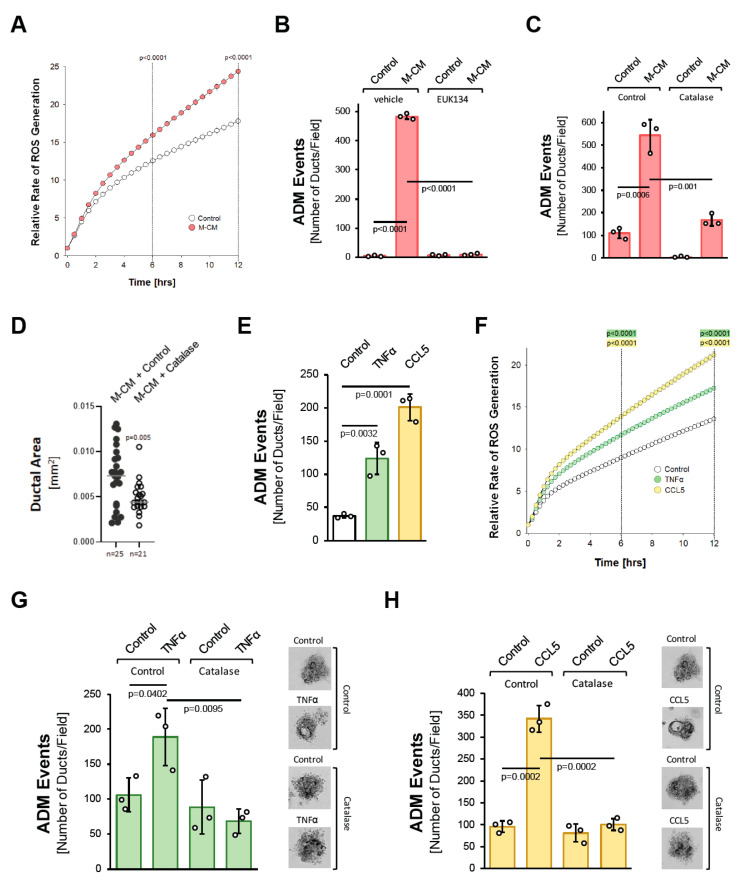
Macrophage-secreted factors induce pancreatic ADM through hydrogen peroxide. (**A**): Primary mouse acinar cells were stimulated as indicated and labeled with H2DFFDA. Generation of intracellular ROS (fluorescent DCF) was measured over the indicated time period. (**B**): Mouse pancreatic acinar cells were isolated and seeded in 3D collagen culture. As indicated, cells were treated with EUK134 (25 µM) and ADM was induced by replacing the overlaying media with macrophage media (Control) or macrophage-conditioned media (M-CM). Ducts formed were quantified as described in the Methods section. (**C**): Mouse pancreatic acinar cells were isolated, infected with rAD-Null or rAD-CVM-Cat and seeded in 3D collagen culture. As indicated, effects on ADM were quantified after stimulation with control media or M-CM. (**D**): Ducts formed after induction of ADM with M-CM following adenoviral expression of catalase (or control) were analyzed for their ductal area using ImageJ. (**E**): Mouse pancreatic acinar cells were isolated, seeded in 3D collagen culture and stimulated with vehicle (Control), TNFα (50 ng/mL) or CCL5 (50 ng/mL). Ducts formed were quantified as indicated in the Methods section. (**F**): Primary mouse acinar cells were stimulated as indicated and labeled with H2DFFDA. Generation of intracellular ROS (fluorescent DCF) was measured over the indicated time period. (**G**,**H**): Mouse pancreatic acinar cells were isolated, infected with rAD-Null or rAD-CVM-Cat and seeded in 3D collagen culture. ADM was induced with 50 ng/mL TNFα (**G**) or 50 ng/mL CCL5 (**H**) and ducts formed were quantified as indicated in the Methods section. Representative pictures are shown on the right side. (**A**–**H**): All experiments shown were performed in triplicates for at least three times and obtained similar results in each repeat. Statistical analysis between two groups was performed using the Student’s *t*-test. A *p* value of 0.05 was considered statistically significant and values are included in the graphs.

**Figure 3 antioxidants-11-00137-f003:**
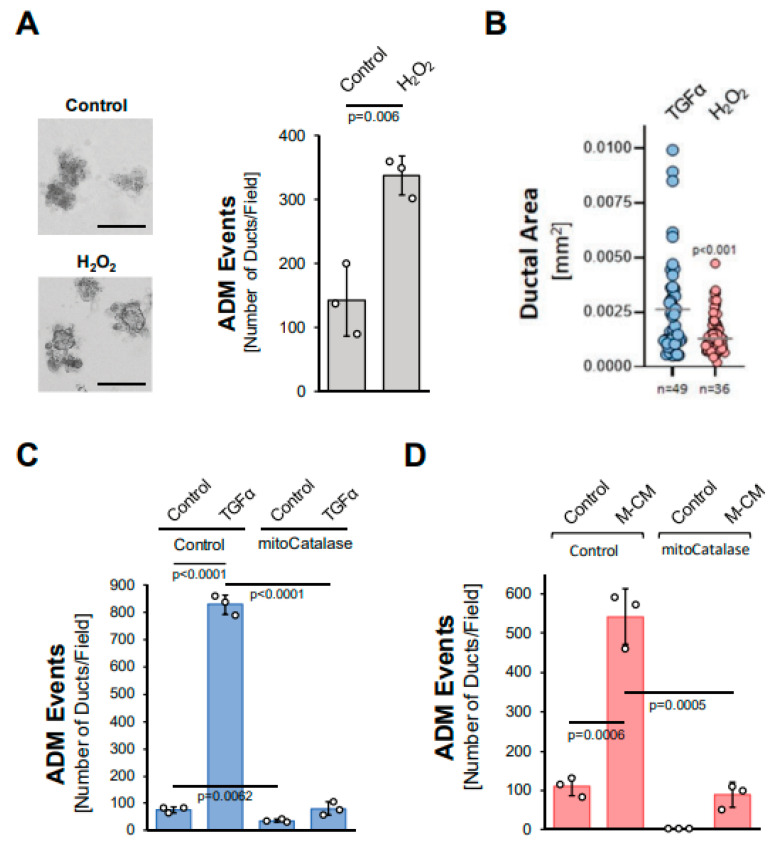
Mitochondria-generated hydrogen peroxide can drive acinar-to-ductal metaplasia. (**A**): Mouse pancreatic acinar cells were isolated, seeded in 3D collagen culture and stimulated with vehicle (Control) or hydrogen peroxide (H_2_O_2_; 10 µM). Ducts formed were photographed (left side; bar = 125 µM) and quantified as indicated in the Methods section. (**B**): Ducts formed after induction of ADM with TGFα (50 ng/mL) or H_2_O_2_ (10 µM) were analyzed for their ductal area using ImageJ. (**C**,**D**): Mouse pancreatic acinar cells were isolated, infected with rAD-Null or rAD-CVM-mCat and seeded in 3D collagen culture. ADM was induced with 50 ng/mL TGFα (**C**) or M-CM (**D**) and ducts formed were quantified as indicated in the Methods section. (**A**–**D**): All experiments shown were performed in triplicates for at least three times and obtained similar results in each replicate. Statistical analysis between two groups was performed using the Student’s *t*-test. A *p* value of 0.05 was considered statistically significant and values are included in the graphs.

**Figure 4 antioxidants-11-00137-f004:**
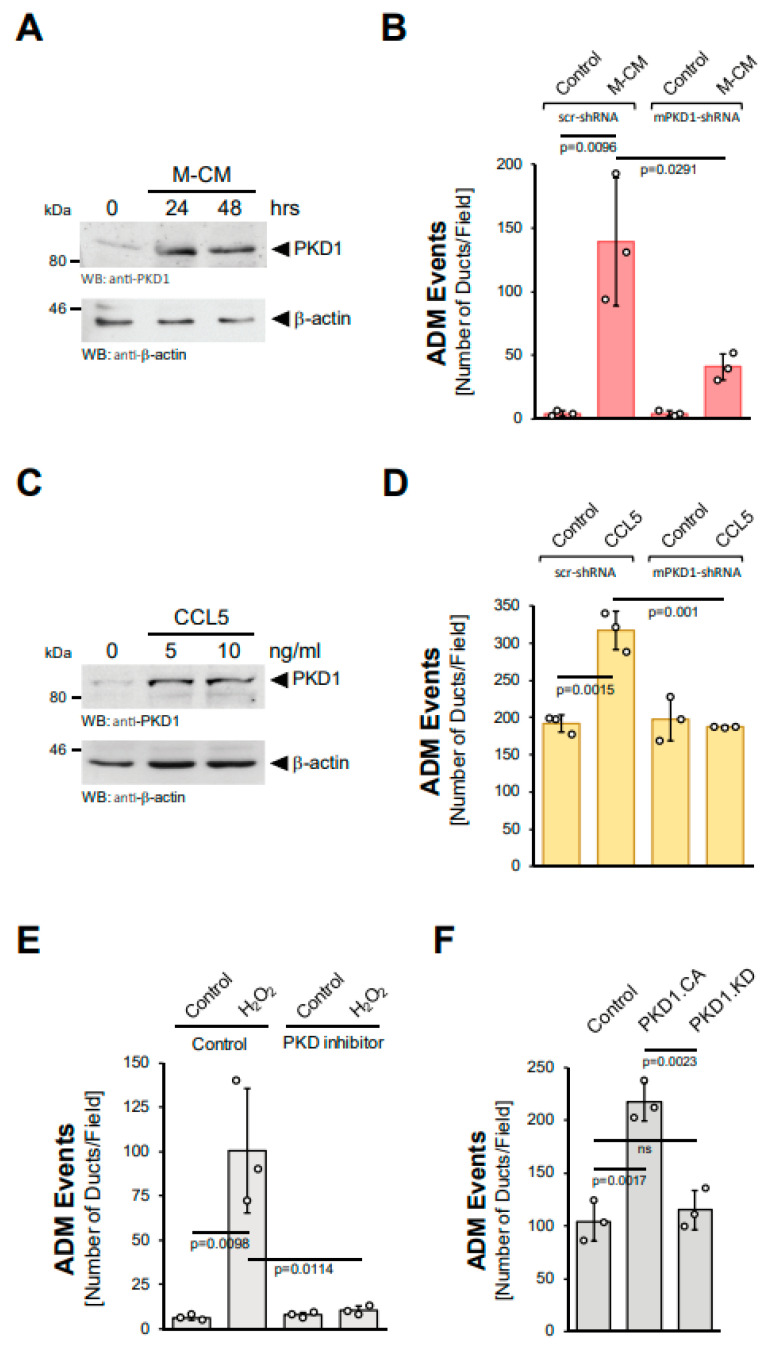
Protein Kinase D1 is downstream of different inducers of ADM. (**A**): Mouse pancreatic acinar cells were isolated and stimulated with M-CM for the indicated time. Cells were lysed and analyzed by Western blot for expression of PKD1 or for β-actin (control for equal loading). (**B**): Mouse pancreatic acinar cells were isolated, lentivirally-infected with control (scr-shRNA) or mPKD1-shRNA and seeded in 3D collagen culture. As indicated, ADM was induced by replacing the overlaying media with macrophage media (control) or macrophage-conditioned media (M-CM). Ducts formed were quantified as described in the Methods section. (**C**): Mouse pancreatic acinar cells were isolated and stimulated with CCL5 at indicated dosages for 48 h. Cells were lysed and analyzed by Western blot for expression of PKD1 or for β-actin (control for equal loading). (**D**): Mouse pancreatic acinar cells were isolated, lentivirally-infected with control (scr-shRNA) or mPKD1-shRNA, seeded in 3D collagen culture and stimulated with vehicle (Control) or CCL5 (50 ng/mL). Ducts formed were quantified as indicated in the Methods section. (**E**): Mouse pancreatic acinar cells were isolated, seeded in 3D collagen culture, treated with vehicle (Control) or PKD inhibitor (kb-NB-142-70, 1 µM) and stimulated with vehicle (Control) or hydrogen peroxide (H_2_O_2_; 10 µM), as indicated. Ducts formed in three replicates were quantified as indicated in the Methods section. (**F**): Mouse pancreatic acinar cells were isolated, lentivirally-infected with either control vector, GFP-tagged PKD1.CA or PKD1.KW and seeded in 3D collagen culture. Ducts formed in three replicates were quantified as indicated in the Methods section. (**A**–**F**): All experiments shown were performed in triplicates for at least three times and obtained similar results in each replicate. Statistical analysis between two groups was performed using the Student’s *t*-test. A *p* value of 0.05 was considered statistically significant and values are included in the graphs.

**Figure 5 antioxidants-11-00137-f005:**
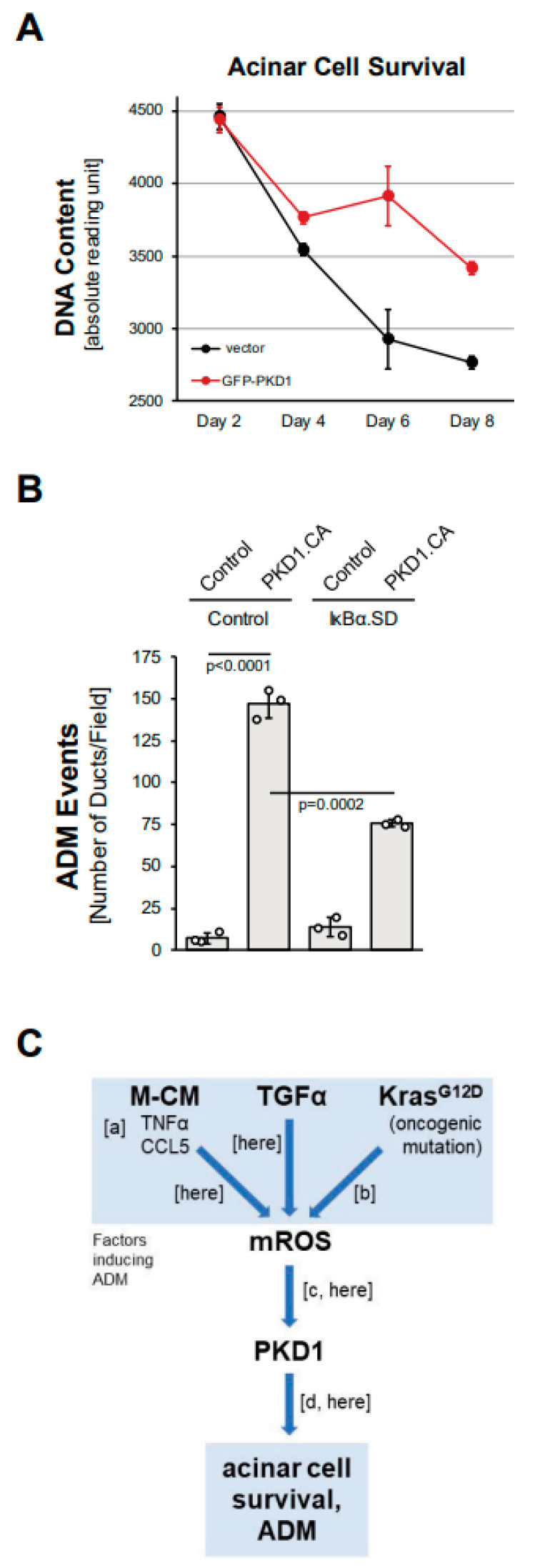
PKD1 increases acinar cell survival and drives ADM through canonical NF-κB signaling. (**A**): Primary mouse acinar cells were isolated, lentivirally-infected with control vector or GFP-tagged PKD1, and then subjected to a CyQUANT Cell Proliferation Assay kit over the indicated time period. (**B**): Mouse pancreatic acinar cells were isolated, adenovirally-infected with control virus or superdominant IκBα (IκBα.SD) and lentivirally-infected with control or GFP-PKD1.CA (GFP-PKD1.SSEE) and then seeded in 3D collagen culture. ADM and ducts formed were quantified as described in the Methods section. (**C**): Schematic summarizing current data and previously published work, demonstrating how ROS-PKD1 signaling is a key factor downstream of different inducers of acinar-to-ductal (ADM) metaplasia. Macrophage-conditioned medium (M-CM) with its two major drivers TNFα and CCL5 (published in [10]; (a)), TGFα (here), or expression of oncogenic KRAS (published in [15]; (b)) all induce mitochondrial reactive oxygen species (mROS), which have been shown to activate PKD1 (published in [17]; [c] and (here)). PKD1 is a key driver of ADM (published in [14]; (d) and (here)).

## Data Availability

Data are contained in the article and Appendix A.

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
