# Peer review of "Generation of Hydrogen Peroxide and Downstream Protein Kinase D1 Signaling Is a Common Feature of Inducers of Pancreatic Acinar-to-Ductal Metaplasia"

_antioxidants, 2022, doi:10.3390/antiox11010137_

Round 1
Reviewer 1 Report
The manuscript is well written with novel findings. Also the introduction, methods and results were arranged properly, discussed accordingly with appropriate conclusion. I accept the manuscript in the current form.
Author Response
We would like to thank this reviewer for stating that our “manuscript is well written with novel findings, and that the introduction, methods and results were arranged properly, discussed accordingly with appropriate conclusion”.
The reviewer indicated that a “English language and style are fine/minor spell check” is required. The revised version of our manuscript was edited by a native speaker (Dr. Alicia Fleming-Martinez; now mentioned in the Acknowledgement section). All changes in the manuscript are indicated in red.
Reviewer 2 Report
The manuscript is well written the presentation of data and content is sound. Suggestion:Maybe at the end of the first sentence in the introduction part a literature citation would be supportive.
In the Material section first paragraph line 69, the company for the beta-actin antibody is missing I assume it is (Sigma?)
Author Response
We would like to thank this reviewer for stating that our “manuscript is well written”, and “the presentation of data and content is sound”. The reviewer had two points that needed to be addressed.
Suggestion: Maybe at the end of the first sentence in the introduction part a literature citation would be supportive – We thank the reviewer for this point and have added a reference in our revised version.
In the Material section first paragraph line 69, the company for the beta-actin antibody is missing I assume it is (Sigma?) - We apologize that we had missed this and now have included this information in our revised version (line 70). It reads: “the β-actin (A5441) antibody was from Sigma-Aldrich (St. Louis, MO).”
Reviewer 3 Report
In my opinion this interesting and well prepared paper before acceptance needs several modifications/corrections. My individual comments are listed below.
The title should be written with first capital letters.
7-9 – The e-mail addresses and authors’ initials should be added.
58-60 – The aim of this study should be reported in better way.
2.4 & 2.5 – The permit of the local ethical commission for animal experiments should be mentioned.
103 – The centrifugation should be characterized by “x g”.
111 – The condition of digestion with collagenase should be reported.
134 – How ROS generation was expressed?
149 – It should be “buffer A”.
152 – The centrifugation must be characterized by “x g” instead of “rpm”.
153 – The condition of SDS-PAGE should be reported.
164 – It should be “Student’s t-test”.
224 – It should be “catalase”.
The list of the abbreviations and Conclusions should be completed at the end of this paper.
454, 457 – It should be “PLoS ONE”.
522/523 – Remove capital letters.
Author Response
We would like to thank this reviewer for describing our manuscript as “interesting and well prepared”. This reviewer had identified some “modifications and corrections”, which we have addressed in full in the following point-to point response.
All changes in the manuscript are indicated in ‘red’.
English language and style are fine/minor spell check required – The revised version of our manuscript was edited by a native speaker (Dr. Alicia Fleming-Martinez; now mentioned in the Acknowledgement section).
The title should be written with first capital letters – We have made these changes in our revised manuscript.
7-9 – The e-mail addresses and authors’ initials should be added – In the revised version of our manuscript we now added all this information.
58-60 – The aim of this study should be reported in better way – We have changed this paragraph in the revised manuscript. It now reads: ‘The aim of the current study was to determine if generation of hydrogen peroxide is a critical factor that is common for different activators of ADM. Another aim was to determine if H2O2 drives the dedifferentiation process to duct-like cells through the kinase PKD1’.
2.4 & 2.5 – The permit of the local ethical commission for animal experiments should be mentioned – We now have added the appropriate information in both sections.
103 – The centrifugation should be characterized by “x g” – We have included this information in line 104.
111 – The condition of digestion with collagenase should be reported – This information now is included in the revised manuscript in lines 114/115.
134 – How ROS generation was expressed? – This information now is included in the revised manuscript in lines in paragraph 2.7.
149 – It should be “buffer A” – We have changed this in the revised version of our manuscript. See line 156.
152 – The centrifugation must be characterized by “x g” instead of “rpm” – We have made these changes in the current revised version in lines 119 and 159.
153 – The condition of SDS-PAGE should be reported – We have included this information in line 161 of our revised manuscript.
164 – It should be “Student’s t-test” – We have made this change in the current revised version in line 171, and in the figure legends.
224 – It should be “catalase” – We have made this change in the current revised version.
The list of the abbreviations and Conclusions should be completed at the end of this paper – In the revision of our manuscript we now included a paragraph named ABBREVIATIONS in lines 394-399 as well as a “5. Conclusions” chapter (lines 384-392). It reads: “Pancreatic acinar-to-ductal metaplasia is a de-differentiation process, and cells that underwent ADM can initiate the formation of pancreatic precancerous lesions when an oncogenic mutation of KRAS is present (reviewed in [1,5]). Multiple inducers of ADM have been identified [7-11], but so far, no common underlying signaling mechanism had been described. We here identify the generation of mitochondrial ROS and ROS-mediated activation of PKD1 as a common signaling pathway for inducers of ADM including CCL5, TNF, EGFR and KRAS. Since PKD1 links to most transcription factors crucial for the acinar cell de-differentiation process, this kinase may be a critical and targetable component to develop strategies to prevent the development of pancreatic cancer.”
454, 457 – It should be “PLoS ONE” – We have made this change in the current revised version.
522/523 – Remove capital letters – We have made this change in the current revised version.